# The Oral Administration of *Sanguisorba officinalis* Extract Improves Physical Performance through LDHA Modulation

**DOI:** 10.3390/molecules26061579

**Published:** 2021-03-12

**Authors:** Jung Ho Han, MinJeong Kim, Hee-Jin Choi, Jung Sook Jin, Syng-Ook Lee, Sung-Jin Bae, Dongryeol Ryu, Ki-Tae Ha

**Affiliations:** 1Department of Korean Medical Science, School of Korean Medicine, Pusan National University, Yangsan, Gyeongnam 50612, Korea; hanjh1013@pusan.ac.kr (J.H.H.); choih@musc.edu (H.-J.C.); 2Healthy Aging Korean Medical Research Center, Pusan National University, Yangsan, Gyeongnam 50612, Korea; jinpaldook@hanmail.net (J.S.J.); dr.nowornever@pusan.ac.kr (S.-J.B.); 3Department of Molecular Cell Biology, Biomedical Institute for Convergence at SKKU (BICS), School of Medicine, Sungkyunkwan University (SKKU), Suwon, Gyeonggi-do 16419, Korea; alswjd0105@skku.edu; 4Department of Food Science and Technology, Keimyung University, Daegu 42601, Korea; synglee@kmu.ac.kr

**Keywords:** *Sanguisorba officinalis* L., herbal medicine, physical performance, LDHA, glycolysis

## Abstract

Muscle fatigue is induced by an acute or chronic physical performance inability after excessive physical activity often associated with lactate accumulation, the end-product of glycolysis. In this study, the water-extracted roots of *Sanguisorba officinalis* L., a herbal medicine traditionally used for inflammation and diarrhea, reduced the activities of lactate dehydrogenase A (LDHA) in in vitro enzyme assay myoblast C2C12 cells and murine muscle tissue. Physical performance measured by a treadmill test was improved in the *S. officinalis*-administrated group. The analysis of mouse serum and tissues showed significant changes in lactate levels. Among the proteins related to energy metabolism-related physical performance, phosphorylated-AMP-activated protein kinase alpha (AMPKα) and peroxisome proliferator-activated receptor-coactivator-1 alpha (PGC-1α) levels were enhanced, whereas the amount of LDHA was suppressed. Therefore, *S. officinalis* might be a candidate for improving physical performance via inhibiting LDHA and glycolysis.

## 1. Introduction

Lactic acid is an important multifactorial biochemical component that plays a key role in the progress of muscular fatigue [1]. Under the hypoxic condition, such as exercise, lactic acid is dissociated into lactate ions and hydrogen (H^+^) entering skeletal myocyte and inducing acidosis, and the accumulation of lactate also negatively affects athletic performance [2]. Besides the normal exercise physiology, lactic acid is a crucial mediator in metabolic diseases reducing muscle performance, such as obesity, type II diabetes, cardiovascular disease, and cancer cachexia [3,4,5].

Lactate dehydrogenase A (LDHA) catalyzed the conversion of pyruvate to lactate by using nicotinamide adenine dinucleotide hydrate (NADH) as a cofactor [6]. Long-period or excessive exercise could increase LDHA activity and result in the accumulation of lactic acid [7]. Several studies have reported that regulating LDHA activity can increase physical performance [8,9]. Thus, reducing the activity of LDHA is expected to improve physical performance. However, in previous studies, serum lactate level or LDHA activity were rarely used for evaluating the effect of medicinal materials for modulating the physical performance, excepting Panax ginseng C.A. Mey and caffeine [10,11].

The *S. officinalis* L., a perennial plant belonging to the Rosaceae family, is widely grown in Northern Hemisphere, including Northern Asia, Northern America, and Europe [12]. *S. officinalis* is an important edible vegetal, and its root is also used in herbal medicine [12,13,14]. The roots of *S. officinalis* have been traditionally used to treat diarrhea, chronic intestinal inflammation, duodenal ulcers, and bleeding hemorrhoids [12]. The ingredient compounds of *S. officinalis*, such as ziyuglycosides I, ziyuglycosides II, sanguiin H-6, ellagic acid, and quercetin, have been reported for various pharmacological action, including antiallergic, anti-inflammatory, anti-obesity, and anticancer [13,15,16]. However, no previous paper reported the effect of *S. officinalis* on the regulation of muscle function and metabolism.

Ongoing screening of LDHA inhibitor from herbal medicine including a spicy nutmeg (fruit of *Myristica fragrans* Houtt.) and its ingredient compound Machilin A [17,18], here we found that *S. officinalis* L. is a potent LDHA inhibitor. Thus, we examined the effect of *S. officinalis* on physical performance and revealed its mechanism of action in the skeletal muscle.

## 2. Materials and Methods

### 2.1. Materials

Antibodies against AMPKα and phospho-AMPK α (p-AMPKα) were purchased from Cell Signaling Technology (Danvers, MA, USA). Antibodies against proliferator-activated receptor gamma coactivator 1-α (PGC-1α), glyceraldehyde 3-phosphate dehydrogenase (GAPDH), heat shock protein 90 (HSP90), and translocase of outer membrane 20 (TOM20) were purchased from Santa Cruz Biotechnology (Santa Cruz, CA, USA). Furthermore, antibodies against CV-ATP5A, CIII-UQCRC2, CII-SDHB, and C1-NDFUB8 were purchased from Invitrogen (Carlsbad, CA, USA). LDHA antibody was supplied by Abcam (Cambridge, MA, USA). The 3-(4,5-dimethylthiazol-2-yl)2,5-diphenyltetrazolium bromide (MTT) were purchased from Merck (Darmstadt, Germany). The gallic acid, catechin, hamamelitannin, ellagic acid, protocatechuic acid, and ziuyuglycoside I was supplied by Sigma-Aldrich (St. Louis, MO, USA).

### 2.2. Extraction of the Roots of S. officinalis

*S. officinalis* was purchased from Omnihber Co. (Daegu, Korea). The *S. officinalis* was harvested in China in late October 2013. The medicinal herb was authenticated by the company’s botanical expert, Daesung Kim. A voucher specimen is kept at the School of Korean Medicine, Pusan National University (HAK-069, Yangsan, Korea). One hundred grams of *S. officinalis* was poured into 1 L of distilled water and boiled for 2 h at 100 °C [19]. Filtration was done with filter paper, and the collected precipitations were discarded. The filtered solution was concentrated with a rotary evaporator and freeze-dried for 1 week. Consequently, the powder was collected, and the weight was measured to calculate the final yield of about 10 g (10%).

### 2.3. High-Performance Liquid Chromatography Analysis

The phytochemical characteristics of *S. officinalis* extract were identified by high-performance liquid chromatography (HPLC) analysis. The HPLC analysis was implemented with an Agilent 1200 series system (Agilent Technologies, Santa Clara, CA, USA), and the LC solution software was used for data analysis. The AkzoNobel (Amsterdam, Netherlands) KR100-5C18 column (4.6 × 250 mm; pore size, 3.5 μm) was used as an analytical column. The solvent used in this study was water containing 0.1% formic acid (solvent A) and methanol (solvent B), and the gradient elution flow was (A)/(B) = 100/0 (5 min) → (A)/(B) = 90/10 (3 min) → (A)/(B) = 70/30 (7 min) → (A)/(B) = 40/60 (10 min) → (A)/(B) = 0/100 (15 min). The analysis was performed at a flow rate of 1 mL/min at the wavelength of 203 and 254 nm detection. The 10 mg/mL of *S. officinalis* extract was injected in 20 μL of distilled water. The 0.2 mg/mL of gallic acid, catechin, hamamelitannin, ellagic acid, protocatechuic acid, and ziyuglycoside I were prepared as standard for ingredient compounds [12].

### 2.4. Mice

Eight-week-old male lab mice (strain C57BL/6 from Hana Bio, Seongnam, Korea) were tested after being quarantined and cared for 1 week after being received. The lighting time was set to 12 h (07:00–19:00), and diet and watering were freely consumed when breeding. A solid feed was provided for experimental animals, and the weight change was measured twice a week. The experimental group was divided into five groups: N—normal, C—control, P—positive control, L—low-dose *S. officinalis*, H; high-dose *S. officinalis*. Five mice were set for each group. Each group was feed with saline solution (group N and C), taurine (500 milligrams of compounds per kilogram of body weight; hereafter, mg/kg, group P), and *S. officinalis* extract (10 or 50 mg/kg, group L and H). Taurine and *S. officinalis* extract were dissolved in a saline solution. The drugs were administered by oral gavage to the mice at 10 AM, three times per week for 4 weeks. All experimental procedures were examined and approved by the Institutional Animal Care and Use Committee of the Southeast Medi-Chem Institute (SEMI; protocol no. SEMI-19-005).

### 2.5. Physical Activity Protocols

#### 2.5.1. Endurance Test

Treadmill running was conducted once a week for 2 weeks during the physical activity adaptation phase and thrice per week for 2 weeks during the physical activity phase. The prepared materials, including *S. officinalis* extract (10 or 50 mg/kg) or taurine (500 mg/kg), were orally administrated to the mice for 4 weeks (thrice per week). The endurance test was assessed after 2 h of fasting. The test was started at a speed of 10 m/min, and the speed was increased by 3 m/min every 3 min until the final speed of 25 m/min was reached. The mice were considered exhausted and scored according to the speed after 10 s, no longer following the treadmill speed. Furthermore, 0, 1, 2, 3, 4, and 5 points were given according to the exhaustion times of 10, 13, 16, 19, 22, and 25 m/min. Consequently, 6 points were given if no exhaustion was noted.

#### 2.5.2. Experimental Mice Sacrifice and Measured Tissue Weight

After the physical activity experiment was completed, the mice were fasted for 16 h, anesthetized with CO_2_, and blood was collected from the abdominal vena cava using a 1 mL syringe. Muscle, liver, kidney, and spleen tissues were washed with physiological saline, and water was removed and measured for each weight.

### 2.6. Tissue Histology

Formalin-fixed paraffin-embedded blocks from the quadriceps muscle were prepared by slicing to 5 μm thickness. These thin slices were stained for hematoxylin and eosin (H&E), as reported previously [20]. Zeiss Axio Imager A1 microscope and Zeiss Axio software (Carl Zeiss, Oberkochen, Germany) were used for images taken in high (×100) magnification areas. Moreover, the muscle fiber area was measured using the ImageJ software.

### 2.7. Assessment of Glycolysis Related Factors

The enzyme-linked immunosorbent assay (ELISA) kit was purchased from MyBioSource (lactate, MBS756195; and creatinine, MBS763433; San Diego, CA, USA). The product of the enzyme–substrate reaction forms a blue-colored complex and measured 450 nm using a microplate reader. The LDH and glucose activities were measured with a commercialized kit (Cobas-c111 Analyzer, Roche, Basel, Switzerland).

### 2.8. Cell Culture

The murine myoblast C2C12 cell lines were generously provided by Helen M. Blau (Stanford University, Stanford, CA, USA). The C2C12 cells were cultured in Dulbecco’s modified Eagle’s medium (Welgene, Seoul, Korea) supplemented with 10% horse serum (Gibco, New York, NY, USA) and antibiotics (100 units/mL penicillin; Invitrogen). The cells were cultured in a humidified CO_2_ incubator at 37 °C, 5% CO_2_ condition. The cells were differentiated into myotubes as previously described [21].

### 2.9. Western Blot Analysis

Either the RIPA or 1% NP-40 lysis buffers containing protease inhibitor cocktail tablets (Roche) were used to extract proteins from murine muscle tissue and C2C12 cells, respectively. Moreover, the buffers were centrifuged for 20 min at 12,000 rpm at 4 °C to eliminate the pellet. After quantifying the protein by the Bradford assay method, the protein was separated by size using sodium dodecyl sulfate–polyacrylamide gel electrophoresis. After transferring the protein to the polyvinylidene fluoride membrane by the transfer system (Bio-Rad, Hercules, CA, USA), the blocking buffer (0.5% skim milk, 1× TBS buffer) containing 5% skim milk was processed for 1 h. The membranes were washed thrice with 1× Tris-buffered saline (TBS) for 10 min and incubated with primary antibodies for proteins at 4 °C overnight. The membranes were then washed thrice with 1× TBS buffer for 10 min. The protein bands were measured with the Western blot detection kit (Bio-Rad), and the chemiluminescence imaging system (ImageQuant LAS 4000; GE Healthcare, Munich, Germany) equipment was used to observe the expression level. Moreover, the membranes were washed thrice with 1× TBS buffer for 10 min. The protein bands were detected using the Western blot detection kit (Bio-Rad) and the chemiluminescence imaging system (GE Healthcare).

### 2.10. LDHA Activity Assay

The reduction of NAD^+^ to NADH was measured according to a previous study [18] to evaluate the LDHA activity. Briefly, the indicated concentrations of *S. officinalis* extract were incubated in a buffer containing 2 mM of pyruvate, 20 μM of NADH, 20 mM of HEPES-K^+^ (pH 7.2), and 10 ng of purified recombinant human LDHA protein for 20 min. The NADH fluorescence with excitation and emission wavelengths of 340 and 460 nm, respectively, were detected using a spectrofluorometer (Spectramax M2; Molecular Devices, Sunnyvale, CA, USA).

### 2.11. Cellular Metabolic Analysis

Extracellular acidification rate (ECAR) and oxygen consumption rate (OCR) were measured using Seahorse XFp analyzer (Agilent Technologies). Moreover, the glycolytic rate and compensatory glycolysis were measured by ECAR. Measurement was started in the basal condition, and oligomycin (1.5 µM), carbonylcyanide *p*-trifluoroumethoxyphenyl hydrazine (FCCP, 0.5 µM), and rotenone (0.5 µM) were sequentially injected at the time of administration. Oxygen concentrations and oxygen consumption rates were measured by OCR, which enables real-time simultaneous measurement of oxygen consumption rate. Oligomycin (1.5 µM) was injected at the basal respiration, and FCCP (0.5 µM) and rotenone (0.5 µM) were sequentially inserted.

### 2.12. Statistical Analysis

The results were calculated and expressed as mean ± standard error of the mean (SEM) from at least three independent experiments. The differences above the mean values of each group were analyzed by Student’s *t*-test, and multi comparisons between groups were analyzed using the analysis of variance with Tukey’s post hoc test by the support of GraphPad Prism software (San Diego, CA, USA).

## 3. Results

### 3.1. The HPLC Analysis of S. officinalis

To validate the phytochemical characteristics of *S. officinalis*, an HPLC analysis was performed using *S. officinalis* extract and standard compounds. The previously known major compounds of *S. officinalis* extract, including gallic acid, catechin, hamamelitannin, and ellagic acid [12,22,23,24], were determined (Figure 1A,B).

### 3.2. The Effect of S. officinalis Extract on Endurance Performance in Mice

To test the effect of *S. officinalis* extract on physical performance, the treadmill test was conducted as summarized in (Figure 2A). The endurance performance of groups P and H improved around 40% compared with group C, whereas group L was not altered in week 3. At week 4, the score indicating the endurance capacity of all groups was elevated compared with group C (Figure 2B). However, body weight did not show any significant difference between each group (Figure 2C). Consequently, the average amounts of water or food intake monitored weekly also showed no significant change (Figure 2D,E). The quadriceps muscle was stained with H&E, and the cross-sectional areas (CSA) were evaluated. Although the mean CSA of group H was slightly increased compared with others, no statistical difference was observed between each group (Figure 2F). These results indicate that the 4-week oral administration of *S. officinalis* extract improved the endurance capacity in mice but did not affect body weight, water and food intake, and muscular structure.

### 3.3. The Effect of S. officinalis Extract on the Blood Biochemistry

Several blood biomarkers, such as serum lactate, LDH, glucose, creatinine, aspartate transaminase (AST), and alanine aminotransferase (ALT), were monitored to reflect physical performance and kidney, liver function. Although the serum lactate and LDH levels were varied, the mean lactate levels of groups C, P, L, and H, which were followed during the 4-week regular physical activity program, were elevated compared with group N, which was not exposed to any physical activity program (Figure 3A,B). The elevated lactate and LDH levels of group C compared with group N indicate that the effect of the last treadmill running was not resolved yet. In line with the improved endurance performance (Figure 3B), the lactate and LDH levels of the groups P, L, and H were lower compared with group C. Lactate, the end-product of glycolysis in the blood is often associated with the blood glucose level. The serum glucose level, which was not significantly changed (Figure 3C), was also measured to verify whether altered blood glucose level contributed to the altered lactate level. Furthermore, serum creatinine, a blood biomarker for kidney toxicity, was also not changed (Figure 3D). No differences were noted in the markers for liver damage (e.g., ALT and AST) among groups (Figure 3E). Liver and kidney weights were measured and significantly appeared only in the liver of group C. However, there are no significant differences were founded in other groups (Appendix A). These results suggest that *S. officinalis* extract could improve physical performance by inhibiting the LDH activity.

### 3.4. The Effect of S. officinalis Extract on AMPK, PGC-1α, and LDHA in Skeletal Muscle

The protein expressions associated with physical performance, such as p-AMPKα, PGC-1α, and LDHA, were examined in skeletal muscle tissues. The phosphorylation of AMPKα was increased in groups P, L, and H, which were given either taurine or *S. officinalis* extract (Figure 4A). The PGC-1α level was also increased in the same groups (Figure 4B). Interestingly, the PGC-1α level of *S. officinalis*-extract-treated groups was almost twice that in group P. The muscle LDHA level of group C was increased compared with group N (Figure 4C). However, the LDHA levels in group P and *S. officinalis*-treated groups (L and H) were reduced compared with group C. Taken together, the oral administration of *S. officinalis* extracts upregulated the p-AMPKα and PGC-1α and downregulated the LDHA in skeletal muscle, which probably contributed to the improvement of physical performance.

### 3.5. The Effect of S. Officinalis Extract on LDHA Activity and Lactate Metabolism in Myocytes

Consistent with in vivo observation, *S. officinalis* extract reduced the amount of LDHA protein in C2C12 myotubes (Figure 5A), while it did not carry out any clear effect on mitochondrial protein, including a subunit of each OXPHOS complex including NDFUB8 (complex I), SDHB (complex II), UQCRC2 (complex III), and ATP5A (complex IV), and the mitochondrial outer membrane protein TOM20 (Figure 5A). Our discoveries showing the improved physical performance (Figure 2B), reduced serum lactate level (Figure 3A), and the reduced LDHA level (Figure 4C) in vivo as well as in vitro led us to ask whether the LDHA activity is also regulated by *S. officinalis* extract. To answer and to understand the mode-of-action of *S. officinalis* extract, we performed an in vitro LDHA activity assay using human-recombinant LDHA protein with *S. officinalis* extract. As expected, in the cell-free LDHA activity assay, *S. officinalis* extracts inhibited the LDHA activity in a dose-dependent manner (Figure 5B). The half-maximal inhibitory concentration of *S. officinalis* extract was estimated to be 17.91 μg/mL. Furthermore, 25 μg/mL of *S. officinalis* extract inhibited almost 60% of LDHA activity to the same extent as a positive control inhibitor, oxamate (20 mM, corresponding to 2.22 mg/mL). Further analysis monitoring ECAR, reflecting lactate production from glycolysis and OCR, and showing mitochondrial respiration indicates that *S. officinalis* extract reduced lactate production without altering mitochondrial respiration (Figure 5C,D). Taken together, these results propose that *S. officinalis* extract-dependent inhibition of LDHA expression and activity regulates the lactate production from glycolysis and improves physical performance (Figure 5E).

## 4. Discussion

In the USA, about 80% of people with chronic fatigue use alternative therapies, including herbal supplements [25]. Especially, muscle injury is a major concern to athletes [26,27]. Both long-duration and high-intensity physical activity cause the accumulation of lactate and subsequently induces the pH imbalance, which consequently generates many free radicals, resulting in damage to muscles [28]. Thus, regulation of the lactate concentration is essential for reducing muscle fatigue and elevating physical performance [29]. To regulate the lactate level, several medicinal herbs have been studied as LDHA potent inhibitors, such as *Pueraria lobata* (Willd.) Owhi, *Belamcanda chinensis* (L.) DC., *Astragalus membranaceus* (Fisch.) Bunge, *Urtica urens* L., *P.* ginseng, *Glycine max* (L.) Merr, *Crocus sativus* L., and *M. fragrans* [18,30,31,32,33]. *P. ginseng* and its ingredient compound ginsenoside Rg3 were reported as anti-fatigue agents via reducing serum lactate levels [11,34]. *M. fragrans* extract also has an anti-sarcopenic effect via regulating IGF1-AKT-mTOR pathway [35]. In this study, it was demonstrated that *S. officinalis* inhibited LDHA activity and increased the physical performance.

LDH is a tetrameric enzyme composed of combining two different isoforms, LDH-M and LDH-H, which are encoded by *LDHA* and *LDHB* genes, respectively [36]. Among five isotypes consisting of a combination of these two subunits, LDH-5 is known as LDHA and abundantly located in skeletal muscle [36,37]. LDHA is the most efficient isoenzyme for catalyzing pyruvate to lactate and regenerating the NAD^+,^ which is necessary for aerobic metabolism in skeletal myocyte mitochondria [38]. As a consequence of the metabolic changes after persistent physical activity, the expression of LDHA was increased in skeletal muscle [39]. Phosphorylation of AMPKα and expression of PGC-1α have been recognized as representative indicators of physical performance in skeletal muscle [40]. Diverse signals activate the AMPK, a sensor of muscle energy change and glycogen levels, via causing its phosphorylation in exercising muscle [41,42]. Phosphorylated AMPK induces the activation of PGC-1α via both expression and phosphorylation levels [43,44]. In addition, the phosphorylation of PGC-1α also affects its own stability and consequently increases the amount of PGC-1α protein as well as its transcriptional activity related to key metabolic pathways, including mitochondrial biogenesis and cellular antioxidant homeostasis [45,46]. Several previous studies revealed the negative correlation between activation of AMPK and PGC-1α and the expression of LDHA [37,47,48]. In this study, our results clearly demonstrated that administration of *S. officinalis* increased the AMPKα phosphorylation and PGC-1α expression and decreased the expression of LDHA in murine skeletal muscle.

Taurine, an essential amino acid synthesized from cysteine and methionine, is widely using as an ingredient of energy drink and exercise supplement. It is well-known that taurine has a beneficial effect on skeletal muscle through the properties of anti-inflammatory, anti-obesity, anti-atherogenic, neuroprotective, and anti-aging [49,50,51,52,53,54]. Especially, taurine enhances physical performance by protecting the damage of muscle cells [55]. Thus, taurine was used as a positive control in this study. Although taurine and *S. officinalis* improved physical performance, the underlying mechanisms of these two agents seem to differ from each other. The effect of taurine on exercise performance is resulted from the reduction of superoxide radical production but does not affect the activity of the antioxidant enzyme [56]. In addition, *S. officinalis* showed a similar extent of the effect on enhancing physical performance at 10 times lesser dose of positive control, taurine. In in vitro LDHA activity assay, the inhibitory effect of 25 μg/mL of *S. officinalis* extract is equivalent to that of positive control, 2.22 mg/mL of oxamate. These results suggest that the potency of *S. officinalis* extract is superior to the positive controls. However, besides to efficacy of medicinal plants, to develop a safe diet therapeutics alleviating muscle fatigue, evaluating the toxicity of each ingredient is very important [57]. In this study, the simple toxicity of *S. officinalis* was measured using the serum levels of creatinine, AST, and ALT, and wights of liver and kidney. The results suggested that *S. officinalis* extract improved physical performance without any renal or hepatic toxicity up to the dose of 50 mg/kg during 4-week oral administration. For more precision safety confirmation, it would need good laboratory practice-level in vivo safety experiments.

Various LDH isoenzymes are expressed in tissue-specific manners, and they have diagnostic values for detecting tissue damage [58]. For example, LDH-3, composed of 2 subunits of LDHA and LDHB, is mainly located in lung tissue, and it is directly correlated to pleural fluid. Thus, it has been proposed as a biomarker of lung tissue damage and pulmonary endothelial cell injury [59]. Especially in muscle damage, which induced by intensive physical performance, LDH-5, composed of 4 LDHA isoforms, is also detected in serum [58]. The use of LDH as a biomarker has been shown to be successfully applied in pharmacokinetic studies [60]. In this study, the serum levels of LDH activity and lactate were clearly decreased by *S. officinalis* administration. Although the precise isoform of LDH in serum was not examined in this study, in the skeletal muscle tissue and C2C12 cells, the expression level of LDHA protein was also diminished. Thus, the further pharmacokinetic study should be conducted to reveal the precise efficacy of *S. officinalis* on the LDH, as a biomarker of muscle damage.

In addition, recent studies of *S. officinalis* revealed that it has several medicinal functions [61,62]. It has several biological functions, such as antioxidant, anti-inflammatory, anti-viral, hemostatic, antimicrobial, anticancer, and anti-Alzheimer disease [63,64,65,66,67,68,69,70]. Among these biological functions, antioxidant, anti-inflammatory, and anticancer are related to the activity of LDHA and/or consequent glycolytic pathway [71,72,73]. Collecting the previous studies and this study, the inhibition of LDHA by *S. officinalis*, might be related to other medicinal functions such as anticancer, antimicrobial as well as improving physical performance.

These findings propose a promising approach to improve physical performance using *S. officinalis* as an inhibitor of LDHA. Indeed, according to traditional use, *S. officinalis* has been used as medicinal supplements in Korea [12]. Compounds contained in the leaves and flowers of *S. officinalis*, such as saponins, vitamin C, and chrysanthemin [12], were reported to enhance physical performance [74,75,76]. In addition, among the compounds of roots of *S. officinalis*, gallic acid and catechin were reported as agents for improving physical performance and enhancing abdominal fat loss, respectively [77,78].

## 5. Conclusions

In this study, it was first showed that *S. officinalis* extract suppresses the activity and expression of LDHA and enhances the physical performance in the mouse. In addition, the AMPK and PGC-1α were activated by administration of *S. officinalis* extract. From these results, we suggest that *S. officinalis* extract may be a potential candidate for developing a health supplement to increase physical performance.

## Figures and Tables

**Figure 1 molecules-26-01579-f001:**
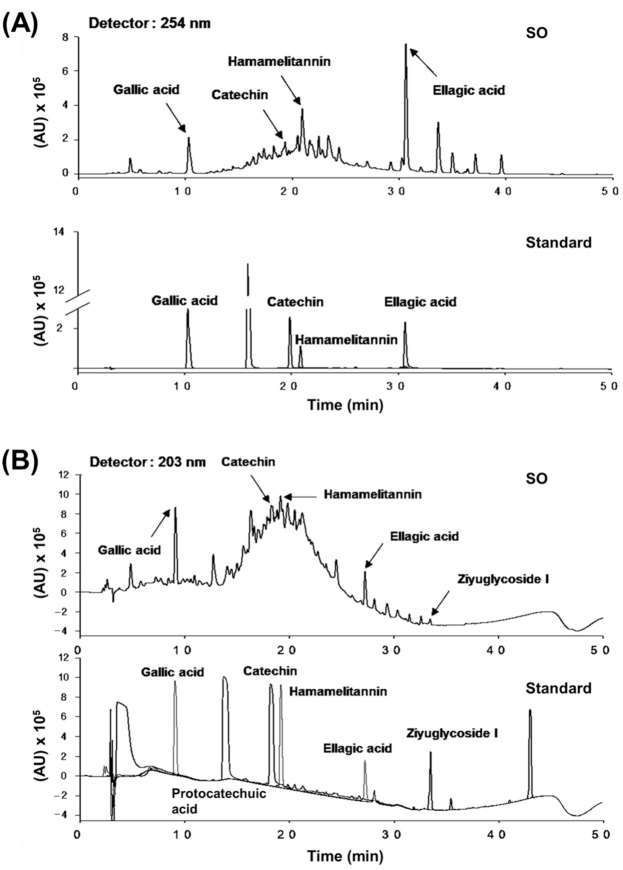
High-performance liquid chromatography (HPLC) analysis of *S. officinalis* (SO) extract. (**A**, **B**) The HPLC chromatogram of *S. officinalis* extract was monitored at ultraviolet detectors of 254 (**A**) and 203 nm (**B**). Representative compounds of *S. officinalis* extract were indicated by arrow lines.

**Figure 2 molecules-26-01579-f002:**
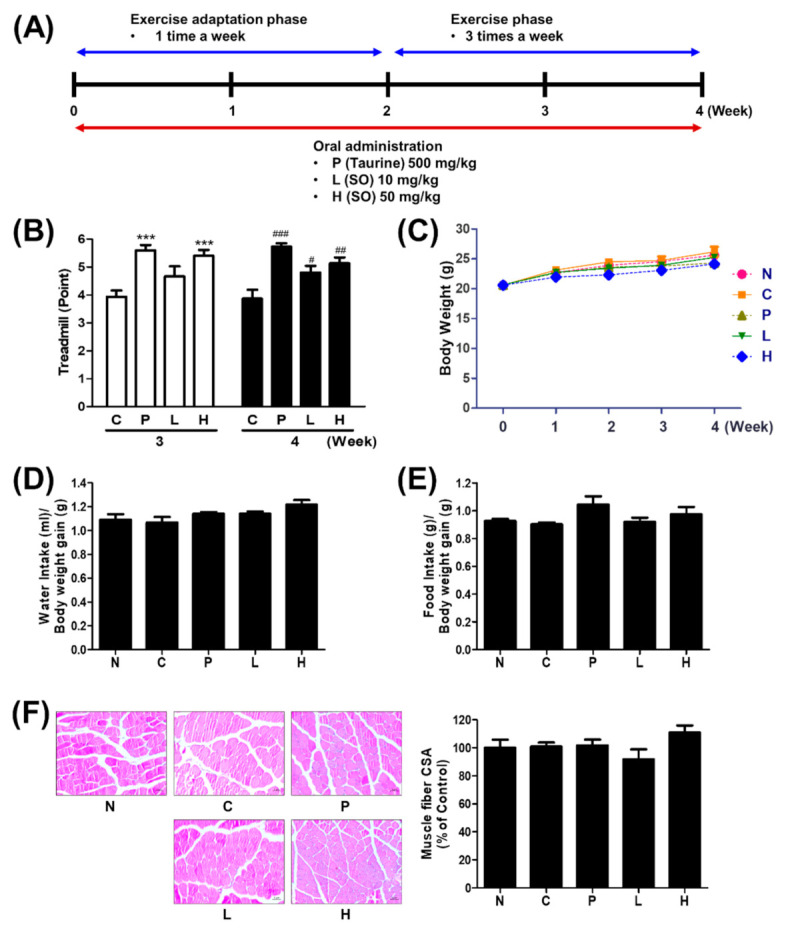
Administration of *S. officinalis* (SO) extract improved physical performance. (**A**) Scheme of treatment and physical activity schedule. (**B**–**D**) The endurance performance (**B**), body weight (**C**), water intake (**D**), and food intake (**E**) of each group were measured. (**F**) Representative H&E staining of the quadriceps muscle tissue. The results of (**B**) are shown as mean ± SEM. *** *p* < 0.001 compared with group C of 3 weeks. # *p* < 0.05, ## *p* < 0.01, and ### *p* < 0.001 compared with group C of 4 week.

**Figure 3 molecules-26-01579-f003:**
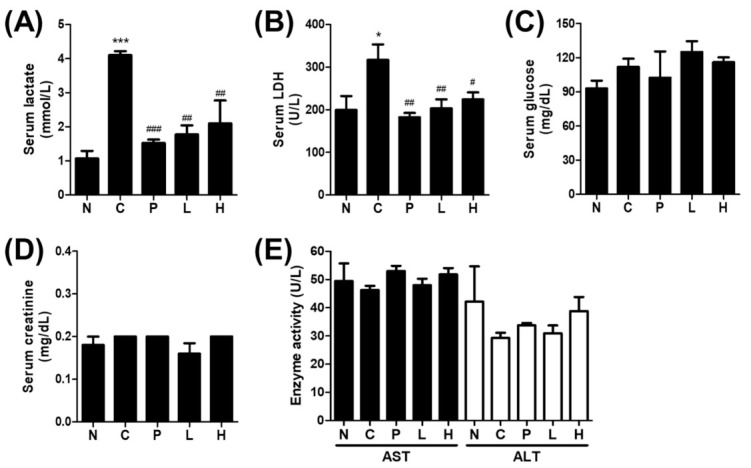
Measurement of physical performance-related factors. Serum levels of lactate (**A**), lactate dehydrogenase (LDH) (**B**), glucose (**C**), creatinine (**D**), and (**E**) aspartate transaminase (AST) and alanine aminotransferase (ALT) were measured by a commercially available assay kit. The results are shown as mean ± SEM. * *p* < 0.05 and *** *p* < 0.001 compared to the N group. # *p* < 0.05, ## *p* < 0.01, and ### *p* < 0.001 compared with group C. N; normal, C—control, P—positive control, L—low-dose *S. officinalis*, H—high-dose *S. officinalis*.

**Figure 4 molecules-26-01579-f004:**
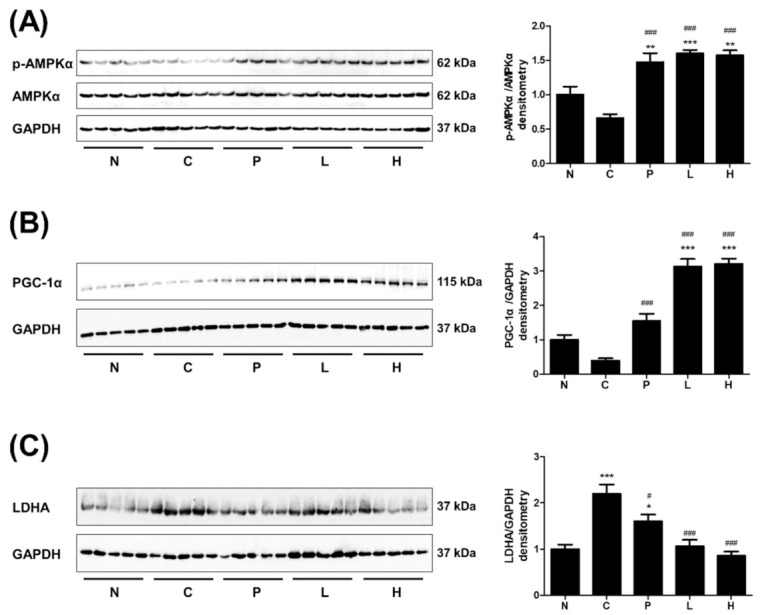
The administration of *S. officinalis* extract modulates physical performance-related factors in murine skeletal muscle. (**A**–**C**) The expressions of phospho-AMPK α (p-AMPKα), phosphorylated-AMP-activated protein kinase alpha (AMPKα), PGC-1α, and lactate dehydrogenase A (LDHA) were evaluated in the quadriceps muscle tissue. GAPDH and AMPKα expressions were used as control. Densitometry was measured with protein expression. The results are shown as mean ± SEM. * *p* < 0.05, ** *p* < 0.01, and *** *p* < 0.001 compared with group N. # *p* < 0.05 and ### *p* < 0.001 compared with group C. N—normal, C—control, P—positive control, L—low-dose *S. officinalis*, H—high-dose *S. officinalis*.

**Figure 5 molecules-26-01579-f005:**
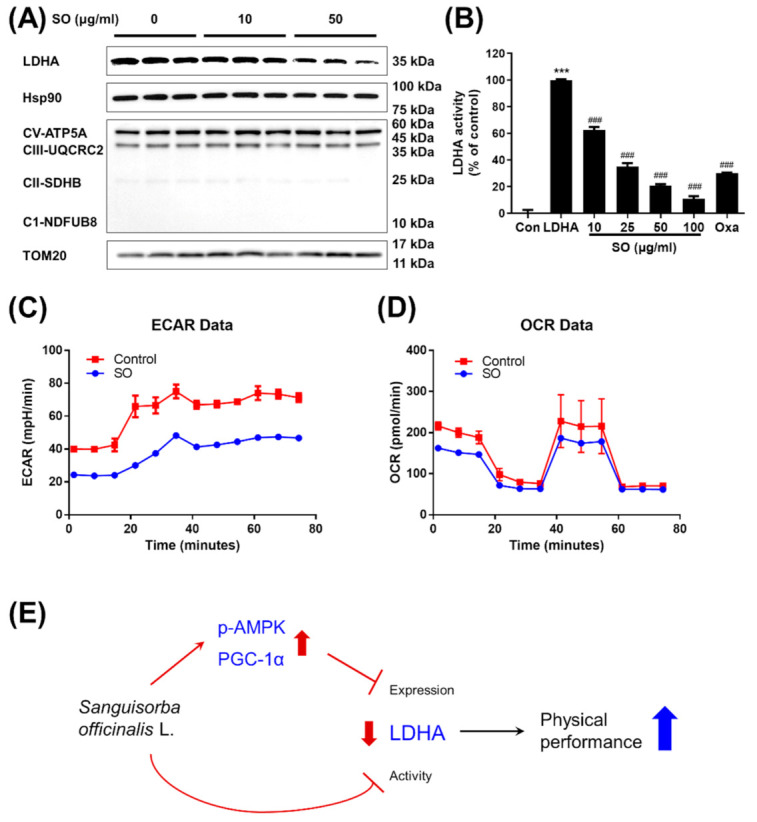
The mode-of-action of *S. officinalis* (SO) (**A**) The protein expression of LDHA, Hsp90, CV-ATP5A, CIII-UQCRC2, CII-SDHB, CI-NDUFB8, and TOM20 were measured by Western blot analysis. Hsp90 expression was used as a control. (**B**) The activity of LDHA was measured by in vitro LDHA assay using recombinant human LDHA in the presence of an indicated concentration of *S. officinalis* extract. The results were shown as mean ± SEM. *** *p* < 0.001 compared to the control. ### *p* < 0.001 compared to the negative control. Analysis of extracellular acidification rate (ECAR) (**C**) and oxygen consumption rate (OCR) (**D**) was performed using a Seahorse XF analyzer to assess glycolysis and mitochondrial respiration, respectively. (**E**) Schematic representation of inhibition of *S. officinalis* extract on the LDHA expression and activity and subsequently enhancing the physical performance was illustrated.

## Data Availability

The data will be made available upon reasonable request.

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
