# Peer review of "The Oral Administration of *Sanguisorba officinalis* Extract Improves Physical Performance through LDHA Modulation"

_molecules, 2021, doi:10.3390/molecules26061579_

Round 1
Reviewer 1 Report
The study of medicinal plants and their medicines for alleviating muscle fatigue is an important aspect. However, while reading the manuscript, I had some questions and comments.
- Specify the name of the plant in accordance with the base http://www.theplantlist.org/
- Please indicate the time of plant collection that you used.
- Indicate the name of the specialist who confirmed the identity of the plant material.
- Indicate the origin of reference compounds (gallic acid, catechin, hamamelitannin, ellagic acid, and ziyuglycoside I).
- To validate qualitative data for reference samples in an extract, provide the regression equations for all compounds (gallic acid, catechin, hamamelitannin, ellagic acid, and ziyuglycoside I) and the lower limit of detection or lower limit of quantification if quantified.
- HPLC analysis conditions are not correct for phenolic compounds of the aqueous extract. Please select other conditions or sample preparation.
- Please replace the term "mpk" on mg/kg.
- What vehicle was used for administration to animals.
- Please explain the presence of additional unidentified peaks on the chromatogram of standards (Fig. 1).
- In section 4, in the part of the conclusion, delete the sentences about what else needs to be done (lines 352-359).
- Abstract should contain specific data obtained in your experiment.
-
Sanguisorba officinalis are widely used in medicine (for example (https://doi.org/10.1016/j.jep.2020.113685) and others. Please discuss the data on the widespread.
-
The direct correlation of LDH and its isoenzymes in pleural fluid was suggested as one of the biomarkers of lung tissue damage and pulmonary endothelial cell injury [M Drent et al., European Respiratory Journal 1996 9: 1736-1742]. It was shown that the use LDH as biomarker and to establish a bioassay method for detecting in rats successfully applied for pharmacokinetic studies [https://doi.org/10.3390/md17100577]. Please discuss this data.
Reviewer 2 Report
Manuscript presented for review with title: “The Oral Administration of Sanguisorba officinalis Extract Improves Physical Performance through LDHA Modulation” is really interesting and very important. The experiment was planned very carefully. I’m impress of excellent work made by authors. The Introduction section includes all necessary information about examined objects and problems.
The collected experimental material and used methods do not raise any objections.
All described sub-chapters in Material and method contains all necessary things, methods and protocols.
The discussion section presents a very good comparison of the obtained results with other results available in the data basis.
The obtained conclusions are clear and present with a high scientific value, according to obtained results.
General opinion: I think, that presented manuscript is a very valuable with high scientific level and should be published in presented form in Molecules journal.
Reviewer 3 Report
Sanguisorba officinalis L. is a plant of the Rosaceae family. It is an edible species, but it is also known for having anti-inflammatory, digestive, astringent, haemostatic and tonic properties. In this work by Han et al., the effects of S. officinalis root extracts on lactate dehydrogenase A (LDHA) activity were evaluated using in vitro enzyme assays, C2C12 myoblastic cells and murine muscle tissue.
General comments
Overall the work is well structured and the Materials and Methods section reports several experimental approaches correctly. The experimental results are interesting as they allow to correlate the inhibition of LDHA expression and activity obtained by administering the S. officinalis extract with the possible improvement of physical performance.
However, some aspects need to be addressed to improve the standard of the manuscript.
In detail
Lines 93-94: Change the sentence to make it clearer. For example, like this:
8-week-old male lab mice (strain C57BL/6 from Hana Bio, Seongnam, Korea) were tested after being quarantined and cared for 1 week after being received.
Line 99: Replace saline with saline solution.
Line 101: Replace S. officinalis with S. officinalis extract.
References Section: There is more recent literature than the topics covered in the manuscript. Therefore, too old bibliographic references need to be updated.
Round 2
Reviewer 1 Report
The authors made the necessary corrections. I have no more questions.